# Retroelement-Linked H3K4me1 Histone Tags Uncover Regulatory Evolution Trends of Gene Enhancers and Feature Quickly Evolving Molecular Processes in Human Physiology

**DOI:** 10.3390/cells8101219

**Published:** 2019-10-08

**Authors:** Daniil Nikitin, Nikita Kolosov, Anastasiia Murzina, Karina Pats, Anton Zamyatin, Victor Tkachev, Maxim Sorokin, Philippe Kopylov, Anton Buzdin

**Affiliations:** 1Group for Genomic Analysis of Cell Signaling Systems, Shemyakin-Ovchinnikov Institute of Bioorganic Chemistry, 117997 Moscow, Russia; danya.nikitin.orel@gmail.com; 2Omicsway Corp., Walnut, CA 91789, USA; tkachev@oncobox.com (V.T.); sorokin@oncobox.com (M.S.); 3ITMO University, 195251 Saint-Petersburg, Russia; nikita-kolosov@yandex.ru (N.K.); murzinanastasiia@gmail.com (A.M.); karina.m.pats@gmail.com (K.P.); zamyatin.anton@gmail.com (A.Z.); 4Institute of Personalized Medicine, I.M. Sechenov First Moscow State Medical University, 119991 Moscow, Russia; fjk@inbox.ru

**Keywords:** human genome evolution, histone modifications, H3K4me1, enhancers of transcription, retrotransposons, retroelements, molecular pathways, gene ontology, epigenetics, gene expression regulation

## Abstract

Background: Retroelements (REs) are mobile genetic elements comprising ~40% of human DNA. They can reshape expression patterns of nearby genes by providing various regulatory sequences. The proportion of regulatory sequences held by REs can serve a measure of regulatory evolution rate of the respective genes and molecular pathways. Methods: We calculated RE-linked enrichment scores for individual genes and molecular pathways based on ENCODE project epigenome data for enhancer-specific histone modification H3K4me1 in five human cell lines. We identified consensus groups of molecular processes that are enriched and deficient in RE-linked H3K4me1 regulation. Results: We calculated H3K4me1 RE-linked enrichment scores for 24,070 human genes and 3095 molecular pathways. We ranked genes and pathways and identified those statistically significantly enriched and deficient in H3K4me1 RE-linked regulation. Conclusion: Non-coding RNA genes were statistically significantly enriched by RE-linked H3K4me1 regulatory modules, thus suggesting their high regulatory evolution rate. The processes of gene silencing by small RNAs, DNA metabolism/chromatin structure, sensory perception/neurotransmission and lipids metabolism showed signs of the fastest regulatory evolution, while the slowest processes were connected with immunity, protein ubiquitination/degradation, cell adhesion, migration and interaction, metals metabolism/ion transport, cell death, intracellular signaling pathways.

## 1. Introduction

Transposable elements occupy nearly one-half of human genome [1]. Among them, retroelements (REs) form the most numerous and active class that shaped ~40% of human DNA [2]. REs impact human gene expression by providing functional regulatory regions including enhancers [3,4]. RE-linked enhancers act via recruiting of transcription factor proteins to the enclosed transcription factor binding sites (TFBS) [5,6]. Such enhancer activity of human REs has greatly influenced complex molecular processes such as innate immunity [7] and placentation [8], and RE-linked enhancers are currently recognized as one of the major driving forces of human regulatory evolution [9,10,11].

Deep sequencing-based technologies such as chromatin immunoprecipitation and sequencing (ChIP-seq) enabled whole genome studies of functional enhancer elements [12]. Monomethylation of H3 lysine 4 (H3K4me1) is major epigenetic mark of both active and dormant enhancers [13]. The whole genome profiles of H3K4me1 mark were resolved [14] and are widely used in gene regulation studies [15]. Investigating epigenomic data in an evolutionary context reveals a complex and constantly changing enhancer landscape [16]. RE-driven enhancers could affect animal diversity [17] and give birth to new genes [18] or even new tissues such as mammalian neocortex [19].

Recent studies of RE regulatory impact on the functioning of human molecular pathways [20,21] provide a computational framework to estimate regulatory evolution of major molecular processes by using RE-linked enhancer activity as the marker. The original analytic pipeline called RetroSpect [22] is based on mapping of RE-linked regulatory sites such as transcription factor binding sites (TFBS) in a close neighborhood of gene transcription start sites (TSSs) [20]. TSS-proximal regions are enriched in major cis-acting regulatory elements, and 10-Kb interval proposed for the RetroSpect analysis covers a significant portion of RE-linked enhancers of human genes [23,24]. Comparing the ChIP-seq signal of RE-linked regulatory elements and of all regulatory elements in the 10-kb frame allows for the calculation of absolute and relative gene enrichments by RE-linked regulatory elements, or GRE and NGRE scores, respectively [21].

RetroSpect methodology enables calculation of relative and absolute RE-linked regulatory enrichment scores for both individual genes and intracellular molecular pathways, i.e. gene ensembles having a common molecular function [25]. After scoring individual genes according to the extent of their RE-linked regulation, the differential genes are then functionally annotated [26]. This annotation can be based on the calculation of molecular pathway activation scores as for gene expression studies [27], or on pathway instability metrics as for DNA mutation data [28]. In the RetroSpect pipeline, absolute and relative RE-linked regulatory enrichment scores for a given molecular pathway are calculated (PII and NPII scores, accordingly), which measure the strength of RE regulation in this pathway [21]. Data on topological structure of few thousands human molecular pathways are stored in public databases such as KEGG [29] and Reactome [30].

Alternatively, the gene set enrichment analysis can be applied which provides an independent way of biological annotation, e.g., based on Gene Ontology (GO) terms enrichment analysis as for the previous application of RetroSpect [31]. Comparison of the results obtained using pathway analysis and GO annotation enhances robustness of the RetroSpect analysis and allows for the reliable identification of consensus molecular processes that are enriched or deficient in RE-linked regulation. These processes, therefore, can be considered as those undergoing accelerated of delayed regulatory evolution [21].

Recent application of RetroSpect to RE-linked regulation of human molecular pathways at the level of TFBS for 13 human cell lines and 563 DNA-binding proteins showed that ~55.5% of human TFBS were connected with REs [21]. Moreover, such molecular processes as gene regulation by microRNAs, olfaction, color vision, fertilization, cellular immune response, and amino acids and fatty acids metabolism and detoxication were found enriched by RE-linked TF regulation [21]. These observed patterns of RE-driven regulatory evolution delineate the most actively evolving—yet different—biological processes [9]. For example, RE-directed evolution of cellular immune response is connected with perpetual evolutionary arms-race between pathogens and hosts [32], whereas color vision is a recent evolutionary innovation of primates [33].

Further examination also evidenced that genes enriched by RE-linked TFBS showed a statistically higher proportion of miRNA and long non-coding RNA (lncRNA) genes than for the set of deficient genes [21].

In this study, we used RetroSpect to investigate RE-linked regulation of human genes by histone modification of H3K4me1, a well-known enhancer mark for the enclosing DNA. We extracted the whole genome of H3K4me1 ChIP-seq profiles for five human cell lines (MCF-7, K562, HepG2, HeLaS3, GM12878) from the ENCODE database. We found that only 27.8% of all H3K4me1 enhancer marks were associated with REs. This is twice lower compared to the previous figure of 55.5% of total RE-linked TFBS [21], which may reflect genome-wide epigenetic repression of RE-linked enhancers.

RetroSpect was then applied to detect their functional impacts. The major molecular processes enriched by RE-linked enhancer regulation were connected with DNA metabolism and maintenance of chromatin structure, sensory perception, neurotransmission and lipids metabolism. The least impacted processes, which may be considered the most conservatively regulated at the level of H3K4me1 [20], dealt with different aspects of the immune response, cell adhesion, migration and interaction, cell death, ion transport and various intracellular signaling pathways.

Moreover, among the top RE-linked regulation enriched genes we found statistically significantly higher proportion of genes coding miRNAs and lncRNAs. At the same time, the proportion of the lncRNA genes was significantly decreased in the bottom cohort of genes sorted by RE-linked H3K4me1 regulation.

## 2. Materials and Methods

### 2.1. Identification of RE-Linked H3K4me1 Modification Tags

Whole genome H3K4me1 ChIP-seq profiles for were extracted from the ENCODE database [34] for five human cell lines (*K562*, *HepG2*, *GM12878*, *MCF-7*, *HeLa-S3*) according to the standard ENCODE histone ChIP-seq protocol [35]. The reference human genome assembly 2009 (hg19) was indexed via Burrows–Wheeler algorithm using BWA software (version 0.7.10) [36]. Concatenation of fastq files with single-end or pairwise reads, alignment to the reference genome and filtering were done using BWA, Samtools (Sanger Institute, Hinxton, Cambridgeshire, UK, version 1.0), Picard (Broad Institute, Cambridge, MA, USA, version 1.92), Bedtools (Quinlan laboratory, University of Utah, UT, USA version 2.17.0), Phantompeakqualtools (Department of Genetics, Department of Computer Science, Stanford University, Stanford, CA, USA, version 1.1) and SPP (Department of Genetics, Department of Computer Science, Stanford University, Stanford, CA, USA, version 1.14) software [36]. Peak calling and signal generation were done using Macs (Harvard University, Cambridge, MA, USA, version 2.1.0) and Bedtools software [37] based on the alignment data. Aligned, filtered and normalized over control ChIP-seq reads for each cell line were mapped on the RE sequences annotated by RepeatMasker (Institute for Systems Biology, Seattle, WA, USA, version 3.2.7) [38] and downloaded from the UCSC Browser (RefGene table) [39]. The list of cell lines investigated here and raw ENCODE data files for each cell line is given in Appendix A.

### 2.2. Gene Expression Data

From the ENCODE database [34] we obtained RNA sequencing gene expression profiles for human cell lines using the following set of filters: “transcription”, “total RNA-Seq” and “gene quantifications”. For three out of five cell lines of interest, we found 19 experiments containing gene expression data in two technical replicates: 11 experiments for *K562* cell line, 5 for *HepG2* and 3 for *GM12878*. Accession numbers are shown in Appendix A.

### 2.3. Measuring Gene Enrichment by RE–Linked H3K4me1 Histone Modification Tags

The coordinates of human genes were downloaded from the USCS Browser (RefGenes table, genome assembly hg19) [39]. For each gene and cell line, all individual REs overlapping with the 10-kb neighborhood of its reference TSS were selected for further analysis. The 10-kb neighborhood covered an interval starting 5 kb upstream and ending 5 kb downstream the TSS. For every known gene in every cell line, we calculated total numbers of H3K4me1 histone modification tags mapped on either RE or RE-free loci in 10-kb neighborhood. We then calculated absolute (GRE) and relative (NGRE) RE-linked regulatory enrichment scores according to the formulas given in Appendix A for 24,070 human genes.

### 2.4. Measuring Molecular Pathway Enrichment by RE– H3K4me1 Tags 

Gene architecture data of the molecular pathways were extracted from the databases BioCarta [40], KEGG [41], NCI [42], Reactome [43] and Pathway Central [44]. Pathways structure data were downloaded in .xml and. biopax formats from these databases and implemented in a computational algorithm Oncobox [21,27,45]. In every cell line, absolute (PII) and relative (NPII) RE-linked regulatory enrichment scores were calculated for 3095 molecular pathways according to the formulas given in Appendix A.

### 2.5. Measuring Enrichment of Gene Sets by Non-Coding RNA Genes

Significance of enrichment or deficiency of proportions of non-coding RNA genes in selected gene sets was evaluated using hyperbolic distribution separately for two classes of non-coding RNA: miRNA and lncRNA. For each gene set we used the following formula of hypergeometric probability mass function to calculate the probability values of numbers higher or lower than observed for miRNA or lncRNA classes:(1)$$p\left(k,M,n,N\right)=\frac{\left(\begin{array}{c} n\\ k \end{array}\right)\left(\begin{array}{c} M-n\\ N-k \end{array}\right)}{\left(\begin{array}{c} M\\ N \end{array}\right)},$$
where *M* is the total number of genes, *N* is the number of non-coding RNA genes in a gene set, *n* is the number of genes in a gene set (e.g., top genes enriched or deficient in RE-linked regulation), *k* is the number of non-coding RNA genes in a sample and *p(k*, *M*, *n*, *N)* is the probability of observation of gene numbers *k*, *M*, *n*, *N*. Brackets denote binomial coefficients. 

For each number of non-coding RNA genes *k* we calculated probability *P(k*, *M*, *n*, *N)* to observe a higher number of non-coding RNA according to the formula:(2)$$P\left(k,M,n,N\right)=\overset{n}{\underset{i=k+1}{\sum }}p\left(i,M,n,N\right),$$
where *p(i*, *M*, *n*, *N)* is the hypergeometric probability mass function defined above.

We used *p* < 0.05 as a significance threshold value for the hypothesis that non-coding RNA genes are overrepresented in a given gene set and *p* > 0.95 for the hypothesis that non-coding RNA genes are not overrepresented. 

Analogously, for each number of non-coding RNA genes *k* we calculated probability *P’(k*, *M*, *n*, *N)* to observe a not higher number of non-coding RNA according to the formula:
(3)$$P^{\prime }\left(k,M,n,N\right)=\overset{k}{\underset{i=1}{\sum }}p\left(i,M,n,N\right),$$
where *p(i*, *M*, *n*, *N)* is the hypergeometric probability mass function defined above.

We used *p* < 0.05 as a significance threshold value for the hypothesis that non-coding RNA genes are underrepresented in a given gene set and *p* >0.95 for the hypothesis that non-coding RNA genes are not underrepresented.

### 2.6. Gene Ontology Enrichment Analysis

Gene Ontology (GO) analysis of genes that are enriched or deficient in RE-linked H3K4me1 histone modification tags (RRE-enriched or RRE-deficient genes, respectively) was performed using DAVID (version 6.8) software [46] using human gene IDs extracted from USCS Genome Browser [47]. The p-values specifying the significance of observed GO-terms enrichment were calculated using a modified Fisher’s exact test [48]. The cut-off for p-values significance was set as 0.05. The enrichment values of GO-terms and Annotation Clusters were calculated as fold changes of their occurrences in the sample and in the human genome [48].

### 2.7. Significance of Correlations

The statistical significance of correlations was calculated as a Pearson correlation coefficient with a *p*-value using the python Seaborn (Michael Waskom, Center for Neural Science, NY, USA, version 0.9.0) package [49]. All calculations were carried out using Sklearn (French Institute for Research in Computer Science and Automation, Rocquencourt, France, version 0.21.2) module [50].

### 2.8. Significance of Correlations

To assess the confidence of the observed patterns for RE-impacted functional processes, we generated 500 sets of randomly permutated GRE and NGRE scores across the cell lines tested by randomly rearranging gene names. For each perturbation, we extracted a set of GRE-NGRE distribution-based 1204 top and bottom genes. These gene sets were profiled by DAVID (Laboratory of Human Retrovirology and Immunoinformatics, Applied/Developmental Research Directorate, Frederick National Laboratory for Cancer Research, MD, USA, version 6.8) software [46] and top-100 GO terms were selected for each set by the lowest *p*-value for each random permutation. Finally, we compared the distributions of *p*-values for the top-100 GO terms for the permuted and real gene sets: real RRE-enriched and RRE-deficient genes were respectively compared with the distributions of RRE-enriched and RRE-deficient genes in random permutations.

The overall RetroSpect data analysis pipeline is shown schematically in Appendix A. All computational codes used here are freely available upon request to the authors.

## 3. Results

### 3.1. Genes and Molecular Pathways Impacted by RE-Linked Histone Modification Marks 

In order to measure the impact of REs on gene regulation by H3K4me1 histone modifications we calculated GRE and NGRE scores for 24,070 human genes (Appendix A). These metrics were calculated separately for five cell lines *MCF-7*, *K562*, *HepG2*, *HeLaS3* and *GM12878* representing four different human tissues. Totally, 2,841,853 H3K4me1 histone modification tags were extracted from the ENCODE project repository including 482,894, 1,075,569, 705,632, 339,874 and 237,884, respectively, for the above human cell lines.

We then correlated these H3K4me1 histone modification profiles with the gene expression data obtained for the same cell lines (Figure 1). In this study, we considered RNA sequencing profiles because they are thought to represent the gold standard data in high throughput transcriptomic research [25]. In the ENCODE project repository, six profiles were available for cell line GM12878, 10 for cell line *HepG2* and 22 for cell line *K562*. For the *MCF-7* and *HeLa-s3* cell lines, no RNA sequencing data were available. We observed a trend that H3K4me1 histone modification profiles positively correlated with the gene expression data (Figure 1). The highest correlations were observed for the long non-coding RNA genes (Figure 1D), the lowest was observed for microRNA genes (Figure 1C) and the intermediate pattern was observed for protein coding genes (Figure 1B). The highest correlations were detected between gene expression and histone modification profiles of the same cell lines (Figure 1). This confirmed that the H3K4me1 histone modification tags from the ENCODE database were related to their expected molecular functions.

We then tested whether gene-wise scores of RE-linked regulatory enrichments (GRE and NGRE) are comparable among the cell lines. We calculated pairwise gene-by-gene Pearson correlation coefficients between five cell lines investigated here for GRE (Figure 2A) or NGRE (Figure 2B) scores. 

Gene-based scores showed a high degree of similarity between the cell lines: pairwise Pearson correlation coefficients were between 0.53 and 0.74 for GRE and between 0.68 and 0.83 for NGRE. Similarly, the PII and NPII scores also showed high similarities between the five different cell lines investigated (Figure 2C,D). We therefore concluded that gene-based scores were congruent among the cell lines tested and, for the next steps of the RetroSpect analysis, they were arithmetically averaged gene-by-gene across the cell lines.

We applied simple linear regression to identify genes that were relatively enriched of deficient by RE-linked H3K4me1 tags. To identify RE-linked regulation (RRE) enriched and deficient genes, we built a trend line using the least-square method in the coordinates of gene GRE and NGRE scores. Intercept of linear model was set to zero (Figure 3). The genes having greater relative (NGRE) compared to absolute (GRE) RE-regulation metric were considered RRE-enriched, and vice versa.

For the above linear model, we sampled 5% of genes having biggest distances from the NGRE-GRE trend line. The above and below 5% gene sets were considered, respectively, RRE-enriched and RRE-deficient (listed in Appendix A).

The same approach was also used for the molecular pathway data (Figure 2C,D and Figure 4). We calculated absolute (PII) and normalized (NPII) pathway involvement scores for 3095 molecular pathways separately for five human cell lines (Appendix A). We then deduced pairwise Pearson correlation coefficients between the cell lines (Figure 2C,D respectively, for PII and NPII). Correlation coefficients for PII varied from 0.53 till 0.66, for NPII they varied from 0.44 till 0.69. Therefore, PII and NPII values were highly congruent between the cell lines investigated and for further analysis we used their averaged values across the five cell lines. 

Similar to the gene-based analysis, we built a linear model and sampled the top and bottom 5% molecular pathways with the biggest distances from the trend line and that were also considered RRE-enriched and RRE-deficient molecular pathways, accordingly (listed in Appendix A).

### 3.2. Functional Characteristics of Top RRE-Enriched and Deficient Genes

In order to functionally characterize RRE-enriched and deficient genes, we performed gene ontology analysis of the corresponding gene sets using DAVID software. We extracted all GO terms corresponding to molecular function, biological process and cellular component [51] and manually sorted them according to their biological roles.

#### 3.2.1. RRE-Enriched Genes

For the RRE-enriched set of human genes (Appendix A) we defined nine major groups of the corresponding GO terms obtained using DAVID software (Appendix A). In toal, there were 32 GO terms with a *p*-value less than 0.05. Among them, six terms were connected with RNA synthesis and degradation, four were connected with DNA- and chromatin-linked processes, three were connected with translation, three were connected with metabolism of lipids, three were connected with metals metabolism and ion transport, two were connected with immune system, two were connected with sensory perception and neurotransmission, and two were connected with the cell cycle regulation. Finally, the last group called “Other and General terms” contained seven terms which solely represented particular biological functions. 

#### 3.2.2. RRE-Deficient Genes

For the RRE-deficient genes (Appendix A), we identified 16 groups of GO terms that passed the statistical threshold (Appendix A). This reflected a bigger number of p-value-filtered GO terms identified (169). Among them, 45 terms were connected with morphogenesis, 23 were connected with RNA synthesis and degradation, 15 were connected with different types of molecular signaling, four were connected with hormone signaling pathways, 11 were connected with programmed cell death, eight were connected with immunity, two were connected with protein ubiquitination and degradation, seven were connected with protein aggregation and import, 12 were connected with cell adhesion, migration and interaction, six were connected with cell cycle and mitosis regulation, four were connected with transport of ions, three were connected with DNA- and chromatin-linked processes, two were connected with response to stress, and two were connected with intracellular processes of response to viruses. Finally, the “Other and general terms” group contained eighteen different terms (Table 2).

#### 3.2.3. Alternative GO Annotation Analysis

Alternatively, we also did functional annotation of RRE-enriched (Appendix A) and deficient (Appendix A) gene sets using Gorilla software and “Gene Ontology Biological Process” database [52]. The only major biological process identified by Gorilla software analysis for the RRE-enriched gene set was gene silencing by microRNA and RNA catabolism; in turn, for the RRE-deficient gene set, (i) protein ubiquitination and (ii) negative regulation of TORC1 signaling processes were identified.

#### 3.2.4. Enrichment by Non-Coding RNA Genes

MicroRNAs and lncRNAs are regulatory non-coding RNA molecules that can specifically modulate gene expression. We evaluated the contents of microRNA and lncRNA genes among the RRE-enriched and deficient genes. We used hypergeometric sampling model to calculate the degree of improbability to obtain the observed numbers of miRNA and lncRNA genes in the top and bottom gene sets by random sampling, summarized on Table 1.

Our findings suggested that both microRNA and lncRNA genes were strongly overrepresented in the group of RRE-enriched genes (*p*-values 2.22e-08 and 3.49e-08, respectively). Moreover, lncRNA genes were also underrepresented in the RRE-deficient cohort (*p*-value 0.0049), which was not the case for the microRNA genes (*p*-value 0.33).

#### 3.2.5. Significance of RRE-Based Functional Gene Annotations

In order to evaluate the significance of the observed annotations obtained using RRE characteristics, we generated 500 random permutations of gene names and corresponding GRE or NGRE scores. For each permutation, 1204 top and 1204 bottom genes were randomly taken and used as described above for the real RRE-top and bottom genes. These randomly generated gene sets were then analyzed using DAVID software, and top-100 GO terms were selected by the lowest *p*-value for each random permutation. Finally, we compared the distributions of p-values for the real and random gene sets (Figure 5A,B for RRE-enriched and deficient genes, respectively). This type of analysis showed that the RRE*-*deficient molecular processes identified here were mostly statistically significant, because the peak value of real distribution (mode) was lower than the mode of random distribution (Figure 5B). However, most of all RRE*-*enriched GO terms overlapped with the random distribution (Figure 5A), although there were few GO terms with significantly lower p-values than for the random permutations. Since none of the 500 random permutations generated GO-terms with p-values lower than those observed for the real RRE-enriched or RRE-deficient genes, the overall q-values of confidence for both groups were smaller than 0.002, thus indicating a high confidence level of the biological processes identified.

### 3.3. Functional Analysis of Top RRE-Enriched and Deficient Molecular Pathways

In order to functionally characterize top RRE-enriched and deficient molecular pathways, we manually classified them into groups according to their biological functions.

#### 3.3.1. RRE-Enriched Pathways

For the top 5% (155 pathways) we totally identified 14 functional groups (Appendix A). Thirty-three pathways were classified as general signaling pathways, 22 were connected with cell cycle and mitosis, seven were connected with cell death, 16 were connected with morphogenesis, 12 were connected with lipid metabolism, 13 were connected with cell adhesion, migration and cell-cell interactions, 11 were connected with the immune system, seven were connected with DNA metabolism and chromatin structure, seven were connected with cytoskeleton formation, five were connected with endocytosis, three were connected with RNA synthesis and degradation, two were connected with the metabolism of amino acids and three were connected with sensory perception and neurotransmission. Finally, the group “Other pathways” included 14 stand-alone pathways.

#### 3.3.2. RRE-Deficient Pathways

For the 5% bottom human pathways we identified 17 functional groups (Appendix A). Thirty-eight pathways were referred as general signaling pathways, 22 pathways were connected with the immune system, 13 were connected with apoptotic processes, three were connected with the cell cycle, 15 were connected with cell adhesion, migration and intercellular interactions, five were connected with protein aggregation and import, three were connected with translation, protein export and folding, eight were connected with lipids metabolism, eight were connected with the detoxication of xenobiotics, eight were connected with morphogenesis, six were connected with DNA metabolism and chromatin structure, four were connected with the metabolism of amino acids, three were connected with hormone-mediated signaling, three were connected with the metabolism and transport of metal ions, four were connected with endocytosis, and two were connected with formation of the cytoskeleton. Ten pathways were included in the group “Other pathways”.

### 3.4. Comparison of Gene- and Pathway-Based RRE Data 

We then compared groups of functional processes found to be RRE-enriched or deficient using two types of analyses: based on GO and on molecular pathways (Table 2).

Among the groups that were annotated during both pathway and GO analysis, there were three RRE-enriched groups, seven RRE-deficient groups and five groups with ambiguous trends (four were RRE-enriched according to pathway analysis and RRE-deficient according to GO annotation; contrarily, one group was RRE-enriched by GO annotation and deficient by pathway analysis). Overall, Matthews correlation coefficient for these two analyses was 0.342, thus indicating their moderate convergence. The consensus groups of molecular processes contained the following pathways and regulatory networks.

#### 3.4.1. RRE-Enriched Processes

(1)The “Posttranscriptional silencing by small RNAs” group was identified using Gorilla functional annotation of RRE-enriched genes and included microRNA-mediated gene silencing.(2)The “DNA Metabolism and Chromatin Structure” group included double strand break DNA repair, transcription-coupled DNA repair, DNA strand displacement and chromatin remodeling processes.(3)The “Sensory Perception and Neurotransmission” group consisted of olfactory receptors activity (more specifically, class C3 metabotropic glutamate pheromonic receptors activity) and retinoid cycle in cones, which is responsible for color vision in primates.(4)The “Lipids Metabolism” group contained fatty acid biosynthesis, beta-oxidation and desaturation of fatty acids as well as phospholipase A2 activity and modification of sterols via cytochrome P450.

#### 3.4.2. RRE-Deficient Processes

(1)“Immune System” was a heterogenous group of molecular processes such as dendritic cell chemotaxis and cytokine production, IL-1, IL-3, IL-4, TLR and PD-1 signaling, asthma-related signaling and activation of RAS GTPase in B-cells.(2)The “Protein Ubiquitination and Degradation” group included ubiquitin-ligase activity, K63 polyubiquitin binding (non-degradative signal), protein degradation in proteasomes and autodegradation of E3 ubiquitin-ligase.(3)The “Cell Adhesion, Migration and Interaction” group included tight junction formation, E-cadherin binding, cell interaction with extracellular matrix (hyaluronanglucosaminidase activity, laminin binding and inhibition of matrix metalloproteinases), MMIF-mediated angiogenesis and platelet aggregation processes.(4)The “Metals Metabolism and Ion Transport” group consisted of zinc and calcium ion binding, chloride and potassium channels activity.(5)The “Cell Death” group contained various signaling pathways responsible for activation of apoptosis (p53, MEF2D-mediated apoptosis in T cells and BAD translocation to mitochondria), PTEN-mediated cell cycle arrest leading to apoptosis, caspase cascade and permeabilization of mitochondrial outer membrane.(6)The “General Signaling Pathways” group included wide variety of signaling pathways such as NK-κB, VEGF, EGF, IGF and mTOR signaling.(7)The “Hormone Signaling Pathways” group included steroid hormone mediated signaling pathways mediating response to estrogen and testosterone and PELP1 modulation of estrogen receptor activity.

## 4. Discussion

In this study we analyzed the whole genome enrichment profiles of histone mark H3K4me1 in order to identify the extent of RE-linked enhancer regulation of human genes and molecular pathways. The data for five human cell lines of different tissue origin (*MCF-7*, *K562*, *HepG2*, *HeLaS3*, *GM12878*) were extracted from the ENCODE repository and analyzed according to the RetroSpect methodology [21,22]. We could only work with the cell lines instead of normal tissues due to public availability of high throughput epigenomic data. The cell lines selected represented four different tissues of human body. However, we found that the histone tags highly correlated between the different cell lines, thus suggesting only minor impact of a tissue-specific component on the data.

Enhancers are long distance-acting cis-regulatory elements that play a central role in mediation of cell type- and cell state- specific variation in gene expression patterns [13]. There are more than 1,000,000 enhancers in the human genome [14]. The chromatin landscape of enhancers is complex and includes a variety of epigenetic signatures such as histone modifications [53], TFBS [54], DNaseI hypersensitivity sites [55] and short non-coding eRNA expression [56].

Molecular functions of enhancers are primarily dependent on TFs that cooperatively bind to multiple clustered TFBS. This includes lineage specific TFs and sequence-dependent effectors of signaling pathways, which allows for the integration of intrinsic and external signals in gene expression regulation [57]. Previously, we utilized RetroSpect to investigate RE-linked TFBS impact on human gene expression and found a group of conservative molecular processes that were enriched or deficient in RE-linked TFBS regulation [21].

Nevertheless, TFBS is not an enhancer-specific epigenetic feature [57]. Though H3K4me1 is a mark of both active and dormant enhancers [58,59], it is a broad enhancer-specific “window of opportunities” that defines active enhancers as well as those primed to activation and deactivated ones [13]. All other enhancer chromatin markers, such as H3K27ac, appear in the context of pre-existing H3K4me1 [13,59] and disappear after downregulation of enhancer activity [58]. We therefore used H3K4me1 as the enhancer hallmark to identify genes and molecular pathways whose regulation is enriched or deficient in RE-linked enhancers—which reflects their regulatory evolution.

Here, we found very high number of non-coding RNAs (microRNA and lncRNA) among the genes enriched in RE-linked enhancer regulation. Moreover, “gene silencing by microRNA and RNA degradation” was the most significant RRE-enriched GO term according to Gorilla software annotation. The same pattern was already observed in the previous study of human genes enriched in RE-linked TFBS regulation [21]. This finding confirms the concept of non-coding RNAs as primary mediators of RE-driven regulatory evolution. Indeed, the non-coding RNAs and REs are tightly interconnected “planets” in the human genome “universe” since RE insertions sometimes generate new non-coding RNA genes that have an evolutionary chance of gaining biological function [60]. Moreover, the non-coding RNA is a source of evolutionary regulatory innovations that dates back to the origin of major Metazoan clades, such as Piwi-interacting RNAs that appeared during the emergence of metazoan multicellularity [61]. Additionally, the evolution of epigenetic repression mediated by small non-coding RNAs (RNA interference) and some other nucleic acids based immune mechanisms were driven by REs and retroviruses as an example of host-pathogen evolutionary arms-race [62]. Therefore, our finding of high RE regulatory load on human non-coding RNAs (detected both at the levels of TFBS and H3K4me1) highlights the vision of non-coding RNAs as the major way of RE-driven regulatory evolution and innovation.

Using RetroSpect functional annotation procedure, we extracted consensus groups of molecular processes that are enriched or deficient in RE-linked enhancer regulation. The set of RRE-enriched processes (three groups) was relatively short compared to the figure of eight groups previously identified by TFBS ¬RetroSpect analysis [21]. This lower number of groups can be explained by the lower number of independent data profiles from ENCODE experimental studies that were included in RetroSpect protocol (one profile for each cell line here and 4260 profiles for different cell lines in [21]). Moreover, the different number of groups of processes can be due to a biological difference of the functional marks investigated: REs can impact gene regulation not only via long-distance acting enhancers, but also via TFBS acting proximal to transcription start site [9]. Therefore, RE-linked TFBS regulatory impact could be broader than the enhancer-only impact.

However, the pattern of molecular processes impacted by RE-linked enhancers agrees well with the previous findings. For example, the high RE-linked enhancer enrichment of the “Sensory perception and neurotransmission” group is in accord with the previously reported enrichment of this group by RE-linked TFBS [21]. This conservative high RRE pattern underlines the fast regulatory evolution of sensory perception and nervous system in the mammalian clade. Indeed, most parts of the mammalian forebrain are linked with the sensory system and have been evolving quickly [63]. Lipids metabolism is another enriched group that also showed quick regulatory evolution in RE-linked TFBS study [21]. The accelerated evolution of human lipidome was shown in several independent lipidomic assays, which was connected with accelerated brain evolution in primate lineage, since lipid composition is crucial for proper functioning of CNS [64]. Moreover, positive selection and coevolution of different lipid metabolism enzymes in primates was detected previously [65] in the context of adaptation of energetic balance and endothermic metabolism [66]. The third RRE-enriched group, DNA metabolism and chromatin structure, primarily contained various processes of DNA repair, which is in agreement with the results of the previous TFBS RetroSpect analysis [21]. This can be explained by redundancy and catalytic promiscuity of DNA repair systems [67]. 

On the other hand, the conservative (RRE-deficient) molecular processes identified here are also in some aspects congruent with those identified previously by TFBS analysis [21] and with some general evolutionary trends established by other methods. For example, the “Protein ubiquitination and degradation” group is one of the most conservative core intracellular processes [68]. Similarly, the “General signaling pathways” group was found as a group with relatively low regulatory evolution, also in the RetroSpect TFBS analysis [21]. The conservative state of this group is most likely due to an enormous complexity and interdependence of core intracellular signaling in mammals. Deregulation of these signaling networks can lead to severe proliferative and developmental disorders in complex long-living organisms such as primates [69]. There are however other means of signaling pathway regulatory evolution such as point mutations and duplications/deletions/translocations of pre-existing regulatory elements [70] that can lead to signaling networks complexity growth [71].

The “Immune system” group was identified here as mostly conservative in contrast to contradictory pattern (RRE-enriched cellular immune response versus RRE-deficient cellular mechanisms of antiviral response) detected previously for the RE-linked TFBS [21]. This difference can be a consequence of lower resolution of RetroSpect methodology for H3K4me1 enhancer mark compared to the larger TFBS datasets (moreTFBS data profiles investigated). Interestingly, in this study, we found three conservative groups important for multicellular organisms, “Cell Adhesion, Migration and Interaction”, “Cell Death” and “Hormone Signaling Pathways” are groups that were not found in the TFBS analysis [21]. In the previous RE-linked TFBS study, the impact of these groups was neither RRE-enriched nor deficient. The last group, “Hormone Signaling Pathways”, was previously identified as RRE-enriched only in the pathway analysis [21]. We speculate that RE-linked regulation for these groups of processes can be rather promoter-proximal than long distance-acting. Though further correct statistical validation of this hypothesis is required, the underlying biological reason for this asymmetry can be the fact that enhancers are long distance acting elements that require additional cis-acting sequences (insulators and anchor elements) and trans-acting factors (coactivators, CTCF protein, etc.) for the correct looping and activation of the cognate promoter [72,73]. Therefore, emergence of a new RE-linked enhancer could introduce more noise and perturbation into authentic gene regulatory networks than the emergence of a RE-linked TFBS near a pre-existing promoter. Since the above three groups of processes need to be tightly regulated in order to orchestrate development of multicellular organism, the “noisiest” RE insertions could be selectively eliminated or silenced [74]. Intriguingly, the same pattern was seen for the group “Metals Metabolism and Ion Transport” as well.

Despite the marked convergence between current results and TFBS-based analysis [21], as much as 10 groups of molecular processes that were established in the TFBS analysis have not been observed here, and three new groups (“Endocytosis”, “Morphogenesis” and “Cytoskeleton”) were observed only in the current study. Among groups annotated in both studies, six groups have the same status based on consensus molecular pathways and GO annotation in both studies, five groups have the same status based on single annotation type (at least in one study) and eight groups have different statuses in these two studies. Overall statistics for the two studies are shown in Table 3.

Again, the possible reason for such differences is a lower resolution of RetroSpect for H3K4me1 mark compared to the TFBS datasets. Another possibility is the prevalence of RE-linked promoter-proximal regulation for groups identified in the TFBS study only [21], whereas RE-linked regulation of genes connected with “Endocytosis”, “Morphogenesis” and “Cytoskeleton” groups could be long-distance acting rather than promoter-proximal. 

Additionally, the possible explanation for the existence of eight groups with different RRE-enrichment statuses is multiple opposite trends in RE-driven regulatory evolution for each of these groups. For example, the “Cell Cycle and Mitosis” group could contain some regulatory networks (such as regulation of cell cycle by certain growth factors) that undergo accelerated regulatory evolution (RRE-enriched), whereas other processes of cell division (such as mitotic spindle formation) could be conservative. Though this scenario is speculative, eukaryotic cell cycle machinery contains both conservative and quickly evolving parts [75]. Further studies are needed to clarify the molecular evolution of processes in groups such as “Immunity”, “Metals Metabolism and Ion Transport”, “Hormones Signaling Pathways”, “Amino Acids Metabolism”, “Detoxication of Xenobiotics”, “RNA Synthesis and Degradation”, “Translation”, “Protein Export and Folding”, “Cell Cycle” and “Mitosis”. 

In this study, we worked with the human cell line data instead of data for the normal tissues due to public availability of high-throughput profiles for target histone marks and RNA sequencing data for the former. The five cell lines selected for our analysis represented different tissues of human body. However, we found that the histone tags highly correlated between the different cell lines, thus suggesting only minor impact of a tissue-specific component on the data. Availability of novel epigenetic datasets corresponding to normal human tissues would be extremely desirable for further re-analysis of data using RetroSpect pipeline.

Here, we used RetroSpect methodology to analyze a histone mark H3K4me1 in the context of RE-linked transcriptional regulation and RE-linked regulatory evolution. This evidences that RetroSpect is applicable to the different types of functional genome landmarks. We hope that examination of new functional genomic markers using RetroSpect methodology will help building an integrated model of human genome regulatory evolution. We also suggest that the new directions for RetroSpect applications could be non-human organisms such as the model species mouse, zebrafish and drosophila.

## 5. Conclusions

In this study we investigated the regulatory influence of RE-linked enhancer elements on human molecular processes using the previously developed RetroSpect analytic pipeline. We found that the most quickly evolving molecular processes under the regulatory impact of RE-linked enhancers were connected with posttranscriptional silencing by small RNAs, DNA metabolism and chromatin structure, sensory perception/neurotransmission and lipids metabolism. The most conservative processes were dealing with immunity, protein ubiquitination and degradation, cell adhesion, migration and interaction, metals metabolism/ion transport, cell death, general and hormones signaling pathways. There was a significant enrichment of non-coding RNA groups among the genes enriched in RE-linked enhancer regulation. Our findings open an avenue towards building an integral picture of RE-driven regulatory evolution and understanding the origin of regulatory complexity and innovation in humans.

## Figures and Tables

**Figure 1 cells-08-01219-f001:**
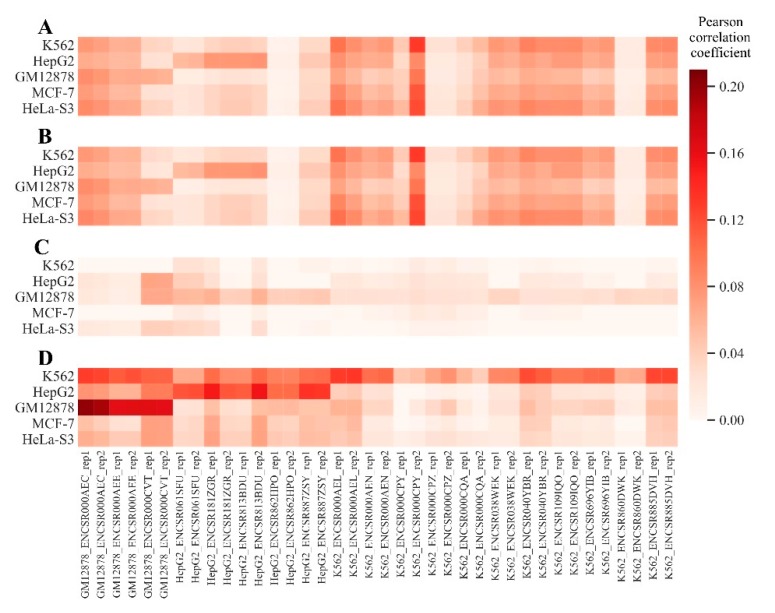
Correlations of H3K4me1 histone modification profiles with the RNA sequencing gene expression data. The histone modification profiles (vertical axis) were compared with normalized gene expressions (abscissa) for 6 *GM12878*, 10 *HepG2* and 22 *K562* cell line RNA sequencing experiments. Pearson correlations were measured for (**A**) set of all known genes; (**B**) set of protein coding genes; (**C**) set of microRNA genes and (**D**) set of long non-coding RNA genes.

**Figure 2 cells-08-01219-f002:**
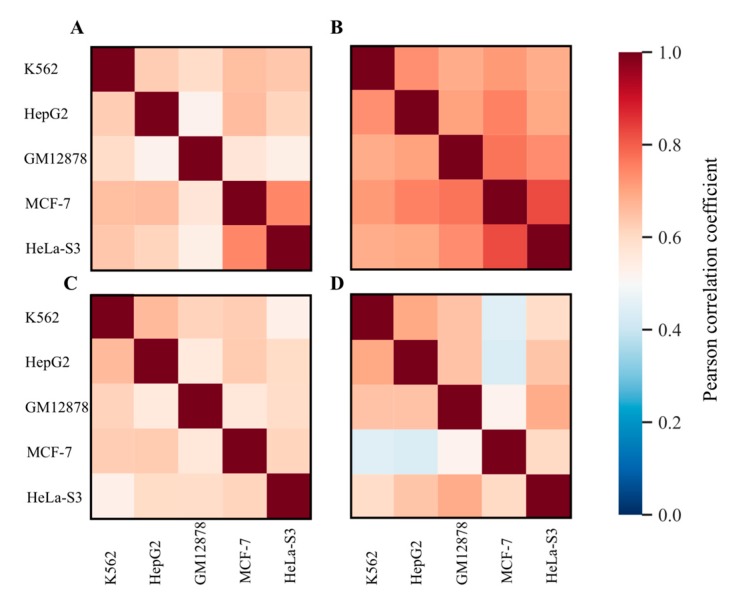
Comparison of RE-linked regulatory enrichment scores between five cell lines investigated. Each panel represents Pearson correlation plot for corresponding score, labels denote cell lines investigated. (**A**) GRE score; (**B**) NGRE score; (**C**) PII score and (**D**) NPII score. Color scale represents Pearson correlation.

**Figure 3 cells-08-01219-f003:**
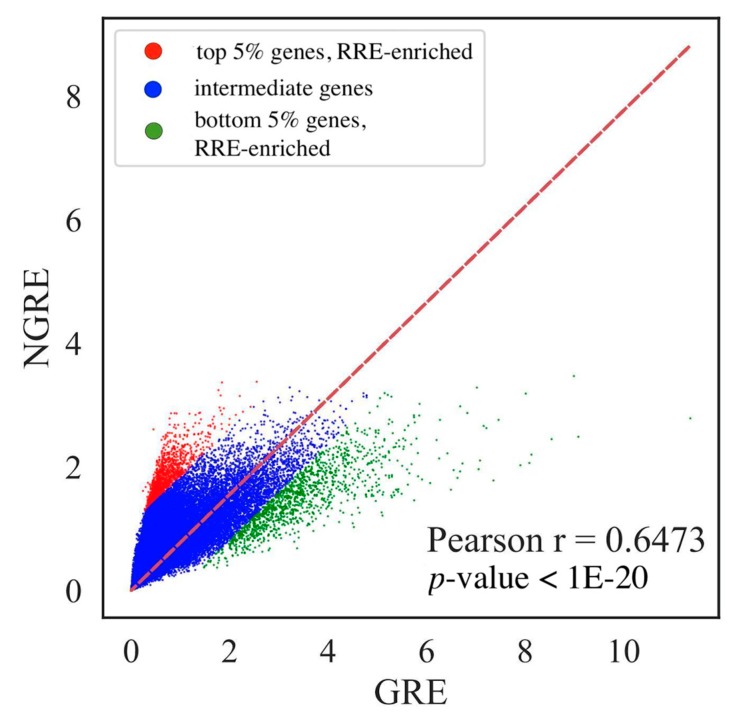
Comparison of GRE (abscissa axis) and NGRE (axis of ordinates) scores for human genes. Color indicates different gene groups (red—RRE-enriched genes, blue—intermediate genes, green—RRE-deficient genes). Each dot represents single gene. Pearson correlation coefficient (r) and Pearson *p*-value (*p*) are shown separately.

**Figure 4 cells-08-01219-f004:**
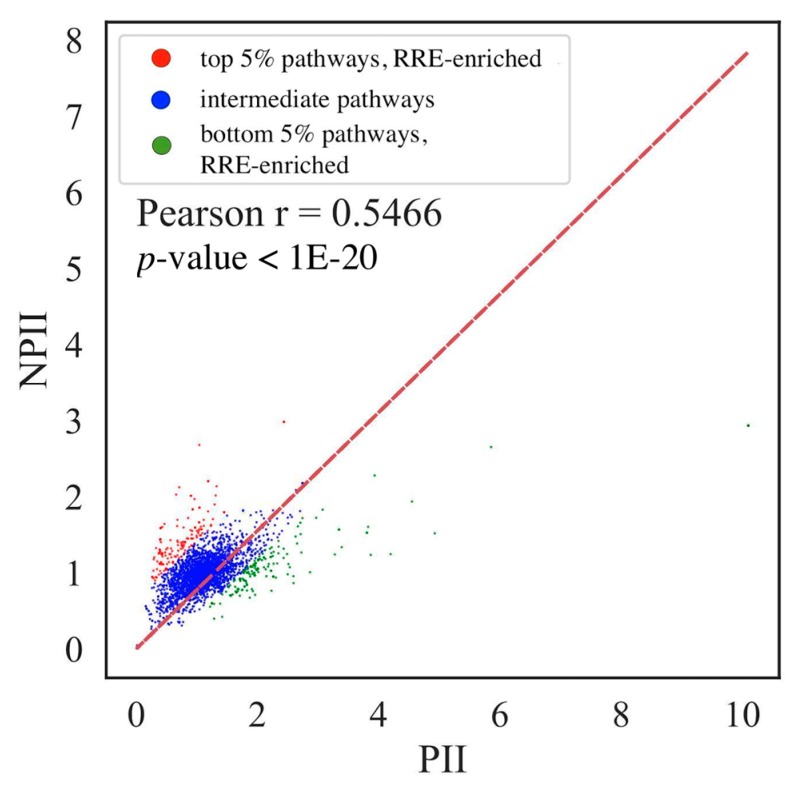
Comparison of PII scores (abscissa axis) and NPII scores (axis of ordinates) for human molecular pathways. Color indicates different pathway groups (red—RRE-enriched molecular pathways, blue—intermediate pathways, green—RRE-deficient molecular pathways). Each dot represents single pathway. Pearson correlation coefficient (r) and Pearson *p*-value (*p*) are shown separately.

**Figure 5 cells-08-01219-f005:**
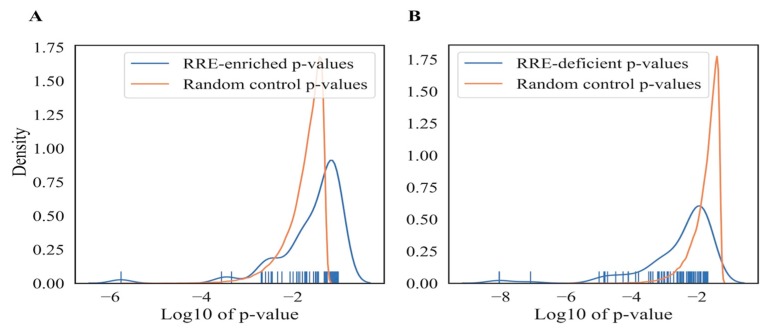
Random control of GO terms enrichment in RRE-enriched (**A**) and RRE*-*deficient (**B**) gene sets.

**Table 1 cells-08-01219-t001:** Analysis of human non-coding RNA RE-linked regulation.

Group	Non-Coding RNA Class	Number of Non-Coding RNA Genes in the Sample	Expected Number of Non-Coding RNA Genes in the Sample, Random Distribution Model	Hypergeometric *p*-Value for Hypothesis That Non-Coding RNA Genes are Overrepresented in the Respective Gene Set	Hypergeometric *p*-Value for Hypothesis That Non-Coding RNA Genes are Underrepresented in the Respective Gene Set	Conclusion
RRE-enriched	lncRNA	145	74	2.22 × 10^−8^	0.99999998	lncRNAs are overrepresented
RRE-deficient	lncRNA	54	74	0.9951	0.0049	lncRNAs are underrepresented
RRE-enriched	microRNA	145	90	3.49 × 10^−8^	0.99999997	microRNAs are overrepresented
RRE-deficient	microRNA	94	90	0.33	0.67	microRNAs are neither overrepresented nor underrepresented

**Table 2 cells-08-01219-t002:** RRE enrichment of the molecular processes according to GO and pathway analyses.

Group of Processes	Pathway Analysis		GO Analysis		Overall Status
	Enriched	Deficient	Enriched	Deficient	
Posttranscriptional silencing by small RNAs	1	0	1	0	enriched
DNA Metabolism and Chromatin Structure	7	6	4	3	enriched
Sensory Perception and Neurotransmission	3	0	2	0	enriched
Lipids Metabolism	12	8	3	0	enriched
Endocytosis	5	4	0	0	enriched
Immune System	11	21	2	8	deficient
Protein Ubiquitination and Degradation	0	5	0	2	deficient
Cell Adhesion, Migration and Interaction	13	15	0	12	deficient
Metals Metabolism and Ion Transport	0	3	3	4	deficient
Cell Death	7	13	0	11	deficient
General Signaling Pathways	33	38	0	15	deficient
Hormones Signaling Pathways	0	3	0	4	deficient
Stress Response	0	0	0	2	deficient
Response to Viruses	0	0	0	2	deficient
Amino Acids Metabolism	2	4	0	0	deficient
Detoxication of Xenobiotics	0	8	0	0	deficient
Protein Aggregation and Import	0	0	0	7	deficient
Morphogenesis	16	9	0	45	ambiguous
Cytoskeleton	7	2	0	7	ambiguous
RNA Synthesis and Degradation	3	0	6	23	ambiguous
Translation, Protein Export and Folding	0	3	3	0	ambiguous
Cell Cycle and Mitosis	22	3	2	6	ambiguous
Other Processes	14	10	7	18	ambiguous

**Table 3 cells-08-01219-t003:** Comparison of isolated groups of processes in current study and in TFBS study [21].

Group of Processes	Current Study	TFBS Study	Comment
Overall Status	Type of Analysis	Status	Type of Analysis
(Pathways/GO/Consensus)	(Pathways/GO/Consensus)
**Consensus Match**	
Sensory Perception and Neurotransmission	enriched	consensus	enriched	consensus	
Lipids Metabolism	enriched	consensus	enriched	consensus	
Protein Ubiquitination and Degradation	deficient	consensus	deficient	consensus	Corresponds to “Translation and Protein Quality Control” in the TFBS study
Posttranscriptional silencing by small RNAs	enriched	consensus	enriched	consensus	Identified by Gorilla software and validated using hypergeometric enrichment in both studies
DNA Metabolism and Chromatin Structure	enriched	consensus	enriched	consensus	Corresponds to “DNA repair” in the TFBS study
General Signaling Pathways	deficient	consensus	deficient	consensus	
**Match by Overall Status**	
Stress Response	deficient	GO	deficient	GO	
Cell Adhesion, Migration and Interaction	deficient	consensus	deficient	GO	
Cell Death	deficient	consensus	deficient	GO	
Protein Aggregation and Import	deficient	GO	deficient	GO	Corresponds to “Protein Localization and Modification” in the TFBS study
Response to Viruses	deficient	GO	deficient	consensus	
**Does not Match**	
Immune System	deficient	consensus	ambiguous	-	
Metals Metabolism and Ion Transport	deficient	consensus	enriched	GO	
Hormones Signaling Pathways	deficient	consensus	enriched	pathways	Corresponds to “Hormones” in the TFBS study
Amino Acids Metabolism	deficient	pathways	enriched	consensus	
Detoxication of Xenobiotics	deficient	pathways	enriched	consensus	
RNA Synthesis and Degradation	ambiguous	-	deficient	GO	
Translation, Protein Export and Folding	ambiguous	-	deficient	consensus	Corresponds to “Translation and Protein Quality Control” in the TFBS study
Cell Cycle and Mitosis	ambiguous	-	deficient	GO	
**Group Appears Only in one of the Studies**	
Endocytosis	enriched	pathways	-		
Morphogenesis	ambiguous	-	-	-	
Cytoskeleton	ambiguous	-	-	-	
Cellular immune response (T cells and NK cells)	-	-	enriched	consensus	
Fertilization	-	-	enriched	consensus	
Vitamin metabolism	-	-	enriched	pathways	
Molecular transport	-	-	enriched	pathways	
Sulfur metabolism and linked redox reactions	-	-	enriched	pathways	
Response to phorbol acetate	-	-	deficient	GO	
Electron transfer reactions	-	-	deficient	GO	
Mitochondria	-	-	deficient	GO	
Nucleic Base, Nucleosides and Nucleotides metabolism	-	-	deficient	consensus	
Carbohydrates metabolism	-	-	ambiguous	-

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
