# Peer review of "Retroelement-Linked H3K4me1 Histone Tags Uncover Regulatory Evolution Trends of Gene Enhancers and Feature Quickly Evolving Molecular Processes in Human Physiology"

_cells, 2019, doi:10.3390/cells8101219_

Round 1

Reviewer 1 Report

Using publicly available database, the authors analyzed whole genome enrichment profiles of histone mark H3K4me1 in order to identify the extent of RE-linked enhancer regulation of human genes and molecular pathways. Data from 5 human cell lines of different tissue origin were arithmetically averaged gene-by-gene across the cell lines. These authors found consensus groups of molecular processes enriched or deficient in RE-linked transcriptional regulation. The findings are interesting and open an avenue towards building an integral picture of RE-driven regulatory evolution.

Author Response

The authors are мукн thankful to Reviewer 1 for his/her thorough reading and positive evaluation of this manuscript

Reviewer 2 Report

Dear authors

you present an original research article about epigenetic enhancer-specific histone methylation in five human cell lines.

Congratulations! This article is well written and the scientifc data is presented very clearly.

The biggest weakness of this study is that it was conducted in five different cell lines from different human tissues. The comparison with direct isolates from human tissue would increase scientific strength of the study, but is not possible in that kind of approach, because epigenomic background data is missing.

So I think it is at the moment not realizable to overcome this weakness for the authors and the study is nevertheless of great scientific value. They could instead compare the genomic data with other cell lines from the same tissue to analyse tissue specific transcriptional regulation, but as the scientific question was a different one in this publications this cannot be accounted for in the suggested revisions.

Strength of the study are in detail:

Authors analysed histon3K4 methylation as a marker for transcriptional regulation in 24 070 genes and over 3000 molecular pathways. These epigenetic modifications were differentiated into retroelement linked and not retroelement linked modifications.

So the study was performed thoroughly.
Signalling pathways enriched in retroelement-linked transcriptional regulation were evolutionary young processes like sensory perception and neurotransmission as well as lipid metabolism. Signalling pathways that lack retroelement linked transcriptional regulation were general signalling pathways and protein degradation, i.e. evolutionary conserved "old" processes.
Even if this was at least partially already known this publications is of scientific impact because of it thorough whole genomic approach.
It delivers the background for future studies on transcriptional dysregulation e.g. in cancer cell lines.

I hope that the authors could provide similar studies in the future in cancer cell lines and in vivo analysis of human tissue and human cancers.

I fully support publication of this article.

Good luck

Author Response

We are very grateful to Reviewer 2 for his/her professional analysis, positive evaluation of this manuscript and useful advice to try focusing on working with the molecular data for the healthy rather than cancer cells in the future. Indeed, as the Reviewer mentioned, so far this kind of data is not publicly available but any further progress with this will open avenue for further validating Retrospect approach.

Reviewer 3 Report

The authors describe the in silico analysis of retroelements (RE) in five human cell lines with regard to their H3K4me1-methylation. They aim to identify genes that may be regulated by H3K4me1-modified RE. They identified gene sets and non-coding RNAs with methylation enriched and deficient REs common to all five cell lines. Finally, they relate biological processes that may be affected by RE-H3K4me1-methylation and compare their study to a tissue factor binding site study. The study was conducted well. However, my major concern with this study is that the authors pool the data from five very different cell lines and do not refer to the differences between them. The cancer-based background of cell lines is not taken into account. In addition, the actual regulation of the determined gene sets was not shown in this – pure in silico – study.

In summary I recommend to accept the manuscript, if the authors can show some experimental data on the regulation of genes and/or non-coding RNAs associated with H3K4me1-methylated REs.

Minor comments:

Figure 1 is misleading. The authors indicate the origin of the cell lines by their tissue. They do not mention the cancer-derived background of the cells.

Line 194: Figure2 C and D are not explained and mentioned later the text without connection.

Lines 204f: what is meant by “relatively enriched of deficient by RE-linked H3K4me1 tags?” In addition, the authors term methylation enriched sites regulated genes. A regulation of these genes by the methylation is likely, but not proven. Please rephrase.

Line 210 and line 228: please enlarge the dots in the legend. The color is barely readable. Please rephrase “middle” genes with a term which describes the status.

Lines 259ff. This analysis is not comprehensible in the current description, please describe the score and its meaning.

I would recommend to exclude figure 7. The content is also presented in table 2.

Author Response

We thank the Reviewer for thorough reading of this manuscript and useful advises. In the revised version, we tried to implement all of them. Below are item-by-item replies to the Reviewer comments.

The authors describe the in silico analysis of retroelements (RE) in five human cell lines with regard to their H3K4me1-methylation. They aim to identify genes that may be regulated by H3K4me1-modified RE. They identified gene sets and non-coding RNAs with methylation enriched and deficient REs common to all five cell lines. Finally, they relate biological processes that may be affected by RE-H3K4me1-methylation and compare their study to a tissue factor binding site study. The study was conducted well. However, my major concern with this study is that the authors pool the data from five very different cell lines and do not refer to the differences between them. The cancer-based background of cell lines is not taken into account. In addition, the actual regulation of the determined gene sets was not shown in this – pure in silico – study.

In summary I recommend to accept the manuscript, if the authors can show some experimental data on the regulation of genes and/or non-coding RNAs associated with H3K4me1-methylated REs.

Answer: We worked with the human cell lines instead of normal tissues due to public availability of high-throughput profiles for target histone marks and RNA sequencing data for the former. The five cell lines selected for our analysis represented different tissues of human body. However, we found that the histone tags highly correlated between the different cell lines, thus suggesting only minor impact of a tissue-specific component on the data. However, availability of novel epigenetic datasets corresponding to normal human tissues would be extremely desirable for further re-analysis of data with RetroSpect pipeline. The Revised version of manuscript was updated with these statements (Discussion sections).

We also added to the revised manuscript comparison of gene expression and H3K4me1 histone modification data that demonstrate the highest congruence for the profiles obtained from the same cell lines (Results section). This evidences functional relevance of the histone modifications data analyzed in this study. A new figure (Fig.1 of the Revised Manuscript) was added illustrating congruence between the epigenetic profiles of the five cell lines investigated and gene expression data. The following subsection was added to Materials and Methods:

“2.2 Gene expression data

From the ENCODE database [34] we obtained RNA sequencing gene expression profiles for human cell lines using the following set of filters: “Transcription”, “total RNA-Seq”, “gene quantifications”. For three out of five cell lines of interest, we found 19 experiments containing gene expression data in two technical replicates: 11 experiments for K562 cell line, 5 - for HepG2 and 3 – for GM12878. Accession numbers are shown in Supplementary Table 1.”

Minor comments:

Figure 1 is misleading. The authors indicate the origin of the cell lines by their tissue. They do not mention the cancer-derived background of the cells.

Answer: We removed Fig.1 from the revised manuscript. To address the cancer background of the cell lines used, the following was added to Discussion section:

“In this study we worked with the human cell line data instead of data for the normal tissues due to public availability of high-throughput profiles for target histone marks and RNA sequencing data for the former. The five cell lines selected for our analysis represented different tissues of human body. However, we found that the histone tags highly correlated between the different cell lines, thus suggesting only minor impact of a tissue-specific component on the data. Availability of novel epigenetic datasets corresponding to normal human tissues would be extremely desirable for further re-analysis of data using RetroSpect pipeline.”

Line 194: Figure2 C and D are not explained and mentioned later the text without connection.

Answer: As suggested by the Reviewer, we referred to panels C and D of Figure 2 in the revised manuscript.

Lines 204f: what is meant by “relatively enriched of deficient by RE-linked H3K4me1 tags?” In addition, the authors term methylation enriched sites regulated genes. A regulation of these genes by the methylation is likely, but not proven. Please rephrase.

Answer: Done. Now it is as follows:

“We applied simple linear regression to identify genes that were relatively enriched of deficient by RE-linked H3K4me1 tags.”

Line 210 and line 228: please enlarge the dots in the legend. The color is barely readable. Please rephrase “middle” genes with a term which describes the status.

Answer: Done. We enlarged dots and replaced “middle” by “intermediate” genes and pathways in figures 3 and 4.

Lines 259ff. This analysis is not comprehensible in the current description, please describe the score and its meaning.

Answer: In the revised manuscript, we explained in detail all major metrics (GRE, NGRE, PII, NPII) and their calculation in Supplementary dataset 2. We hope now it is described appropriately.

I would recommend to exclude figure 7. The content is also presented in table 2.

Answer: We removed Figure 7 as suggested by the Reviewer

Round 2

Reviewer 3 Report

The authors responded to all my concerns adequately.